# Reproduction ratio and growth rates: Measures for an unfolding pandemic

**Maíra Aguiar** [1,2,3]⊛*, **Joseba Bidaurrazaga Van-Dierdonck**[4]⊛, **Nico Stollenwerk**[1,5]⊛

**1** Dipartimento di Matematica, Università degli Studi di Trento, Trento, Italy, **2** Basque Center for Applied Mathematics (BCAM), Bilbao, Spain, **3** Ikerbasque, Basque Foundation for Science, Bilbao, Spain, **4** Public Health, Basque Health Department, Gobierno Vasco, Bilbao, Spain, **5** Center for Mathematics, Fundamental Applications and Operations Research, Lisbon University, Lisbon, Portugal

⊛ These authors contributed equally to this work.
* m.aguiar@unitn.it

**Data Availability Statement:** The data underlying the results presented in the study are available from https://www.euskadi.eus/boletin-de-datos-sobre-la-evolucion-del-coronavirus/web01-a2korona/es/ Epidemiological data used in this

## Abstract

The initial exponential growth rate of an epidemic is an important measure that follows directly from data at hand, commonly used to infer the basic reproduction number. As the growth rates $\lambda(t)$ of tested positive COVID-19 cases have crossed the threshold in many countries, with negative numbers as surrogate for disease transmission deceleration, lockdowns lifting are linked to the behavior of the momentary reproduction numbers $r(t)$, often called $R_0$. Important to note that this concept alone can be easily misinterpreted as it is bound to many internal assumptions of the underlying model and significantly affected by the assumed recovery period. Here we present our experience, as part of the Basque Country Modeling Task Force (BMTF), in monitoring the development of the COVID-19 epidemic, by considering not only the behaviour of $r(t)$ estimated for the new tested positive cases—significantly affected by the increased testing capacities, but also the momentary growth rates for hospitalizations, ICU admissions, deceased and recovered cases, in assisting the Basque Health Managers and the Basque Government during the lockdown lifting measures. Two different data sets, collected and then refined during the COVID-19 responses, are used as an exercise to estimate the momentary growth rates and reproduction numbers over time in the Basque Country, and the implications of using those concepts to make decisions about easing lockdown and relaxing social distancing measures are discussed. These results are potentially helpful for task forces around the globe which are now struggling to provide real scientific advice for health managers and governments while the lockdown measures are relaxed.

## Introduction

As the COVID-19 pandemic is unfolding, research on mathematical modeling became imperative and very influential, not only in understanding the epidemiology of COVID-19 but also

study are provided by the Basque Health
Department and the Basque Health Service
(Osakidetza) to the Basque Modeling Task Force
members, collected with specific inclusion and
exclusion criteria. Epidemiological data used in this
study are provided by the Basque Health
Department and the Basque Health Service
(Osakidetza) to the Basque Modeling Task Force
members, collected with specific inclusion and
exclusion criteria.

**Funding:** Maíra Aguiar has received funding from
the European Union's Horizon 2020 research and
innovation programme under the Marie
Skłodowska-Curie grant agreement No 792494.

**Competing interests:** The authors have declared
that no competing interests exist.

in helping the national health systems to cope with the high demands of hospitalizations, for
example, providing projections and predictions based on the available data. Used as public
health guiding tools to evaluate the impact of intervention measures, governments have
already taken important decisions based on modeling results [1–3].

While for diseases which are long established, such as measles [4], e.g., modeling results are
easier to be interpreted, given the availability of long term and well established data collections,
and public health interventions, counting with an effective vaccine, are able to be be imple-
mented in time to avoid large outbreaks. For COVID-19 the situation is completely different.
We are now dealing with a single disease outbreak in a pandemic scenario and modeling pro-
jections, for instance, need to be adjusted for the new scientific information and new data that
are generated every day under unprecedentedly fast changes of circumstances. Governments
around the globe are relying on quick measurements updates and long established concepts
such as the reproduction numbers which are bound to too many internal assumptions of the
underlying model, smoothing and approximations [5] and significantly affected by the
assumed recovery period.

The initial exponential growth rate of an epidemic is an important measure that follows
directly from data at hand, commonly used to infer the basic reproduction number $R_0$, which
is the number of secondary cases generated from a primary infected case during its infectious-
ness before recovering in a completely susceptible population [6]. Both concepts can be
extended to larger compartmental models and into the phase when effects of the control mea-
sures become visible and parameters slowly change, leading to the so called momentary growth
rates $\lambda(t)$, and momentary reproduction ratios $r(t)$.

In the beginning of COVID-19 epidemics, the process of collecting data were often not yet
well organized or pre-organized in the way that we could immediately use to feed models and
extract accurate measurements for the momentary growth rates and the momentary reproduc-
tion numbers. To mitigate and suppress COVID-19 transmission, draconian intervention
measures were rapidly implemented, crippling our economies as lockdowns were imple-
mented. As research to develop an effective vaccine is ongoing, epidemiologists and public
health workers are the frontline of this pandemic, focusing on the well known public health
surveillance strategies of testing, isolation and contact tracing of infected COVID-19 individu-
als. Up to date, more than 4 million cases were confirmed with about 300 thousand deaths,
and these numbers are still increasing [7].

After several weeks of social distancing restrictions, lockdowns start now to be lifted
and modeling task forces around the globe are struggling to apply the concept of $r(t)$,
often called $R_0$, to decide whether social distancing relaxation decisions are taken in the
right period of time, i.e, when the outbreak is assumed to be controlled, with negative
growth rates and a momentary reproduction numbers below 1. Although the absolute
value of $r(t)$ can vary given the modeling assumptions considered during its estimation, it
is advised to rather look at the threshold behavior, using the growth rates primarily as it is
independent of modeling uncertainties and clearly indicates if the outbreak is under control
or not when estimations are below or above threshold. Complementary measures of growth
rates for different variables such as hospitalization, intensive care units (ICU) admissions
and deceased, where data is also collected, should be evaluated when political decisions are
taken.

In this paper we present the growth rates and reproduction numbers for the COVID-19
epidemic in the Basque Country, an autonomous community in northern Spain with 2.2
million inhabitants. For the reproduction number calculation we use a refined stochastic

SHARUCD-type model—an extension of the well known simple SIR model that is frequently used to model different disease outbreaks [8–10], developed within a multidisciplinary task force (so-called Basque Modelling Task Force, BMTF) created to assist the Basque Health managers and the Basque Government during the COVID-19 responses. The model is calibrated using the empirical data provided by the Basque Health Department and the Basque Health Service (Osakidetza), continually collected with specific inclusion and exclusion criteria. Able to describe well the incidences of disease for different variables of tested positive individuals (see Fig 1), this framework is now used to monitor disease transmission, including estimations of the momentary growth rates and reproduction numbers, while the country lockdown is gradually lifted [11]. Using two different available data sets for the Basque Country, collected from March 4 to May 9, 2020, the data was revised with a change on the variable definition for positive cases in respect to the diagnostic test used. We present results obtained for the momentary growth rates and reproduction ratios during the ongoing COVID-19 epidemic in the Basque Country and discuss the implications of using those concepts during an unfolding pandemic.

## Materials and methods

For the Basque Country we use the cumulative data for the following variables defined as: i) total tested positive patients ($I_{cum}$) which are recorded in categories for ii) hospital admissions ($C_H$), iii) intensive care units admissions ($C_U$), iv) recovered ($C_R$) and v) deceased ($D$). At the beginning of the outbreak, only patients with severe symptoms admitted to a hospital were tested using the PCR (polymerase chain reaction) method. As testing capacities increased, including also antibody tests used mainly as screening tool in nursing homes, less severe symptomatic cases started to be tested, contributing to enhance the number of confirmed positive cases in the population. This data collection (named "Data set A") includes, for each category or variable, patients tested with both PCR and rapid antibody tests. "Data set A" has now being revised to include patiences, in all categories, who were tested positive with PCR method only (named "Data set B"). Using the data for the all positive cases, the momentary growth rates ($\lambda$) and the momentary reproduction numbers ($r(t)$) are calculated for both data sets, A and B, and results are compared.

### 0.1 The underlying mathematical model and empirical data

We use SHARUCD-type models, an extension of the well known simple *SIR* (susceptible-infected-recovered) model, with susceptible class $S$, infected class partitioned into severe infections prone to hospitalization ($H$) and mild, sub-clinical or asymptomatic infections ($A$). For severe infections prone to hospitalization, we assume that individuals could either recover $R$, be admitted to the ICU facilities $U$ or or eventually deceased into class $D$. Since we investigate the cumulative data on the infection classes (and not prevalence), we also include classes $C$ to count cumulatively the new cases for "hospitalized" $C_H$, "asymptomatic" $C_A$, recovered $C_R$ and ICU patients $C_U$. The deceased cases are automatically collecting cumulative cases, since there is no exit transition form the deceased class $D$. For more information, please see also [11]

We consider primarily SHARUCD model versions as stochastic processes in order to compare with the available data which are often noisy and to include population fluctuations, since at times we have relatively low numbers of infected in the various classes. The stochastic version can be formulated through the master equation in application to epidemiology in a

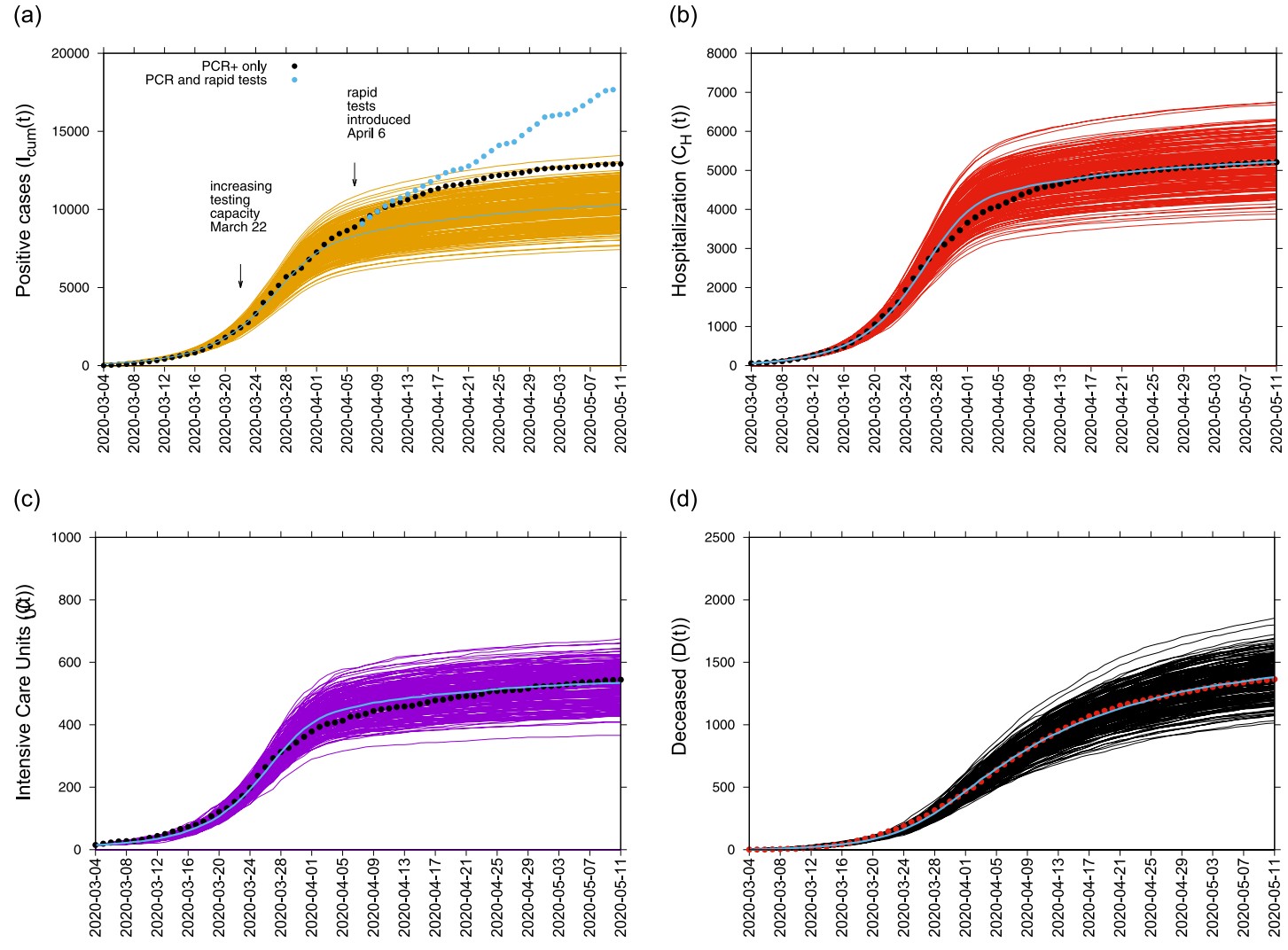

**Fig 1. Ensemble of 200 stochastic realizations of the SHARUCD-type model.** Mean of the stochastic realizations is shown with a blue line. In a) Cumulative tested positive cases $I_{cum}(t)$. Data set From April 6, 2020, we note an increase of reported positive cases as the testing capacities were increasing. In b) cumulative hospitalized cases $C_H(t)$, c) cumulative ICU admission $C_U(t)$, d) cumulative deceased cases $D(t)$. For reference, "Data set A" is plotted as black dots in a-c) and red dots in d). "Data set B" is plotted as blue dots in a). Empirical data was provided by the Basque Health Department and the Basque Health Service (Osakidetza).

generic form using densities of all variables $x_1 := S/N$, $x_2 := H/N$, $x_3 := A/N$, $x_4 := R/N$, $x_5 := U/N$, $x_6 := C_H/N$, $x_7 := C_A/N$, $x_8 := C_U/N$ and $x_9 := D/N$ and $x_{10} := C_R/N$ hence state vector $\underline{x} := (x_1, ..., x_{10})^{tr}$, giving the dynamics for the probabilities $p(\underline{x}, t)$ as

$$\frac{d}{dt} p(\underline{x}, t) = \sum_{j=1}^{n} \Big( N w_j(\underline{x} + \Delta \underline{x}_j) \cdot p(\underline{x} + \Delta \underline{x}_j, t) - N w_j(\underline{x}) \cdot p(\underline{x}, t) \Big) \tag{1}$$

with $n = 10$ different transitions $w_j(\underline{x})$, as described by the mechanisms above, and small deviation from state $\underline{x}$ as $\Delta \underline{x} j := \frac{1}{N} \cdot \underline{r} j$. For the refined SHARUCD model we have explicitly the

following transitions $w_j(\underline{x})$ and its shifting vectors $\underline{r}j$ given by

$$w_1(\underline{x}) = \eta(1 - v)\beta x_1(x_2 + \phi x_3 + \varrho) \qquad , \quad \underline{r}_1 = (1, -1, 0, 0, 0, -1, 0, 0, 0, 0)^{tr}$$

$$w_2(\underline{x}) = \xi(1 - \eta)\beta x_1(x_2 + \phi x_3 + \varrho) \qquad , \quad \underline{r}_2 = (1, 0, -1, 0, 0, 0, -1, 0, 0, 0)^{tr}$$

$$w_3(\underline{x}) = (1 - \xi)(1 - \eta)\beta x_1(x_2 + \phi x_3 + \varrho) \quad , \quad \underline{r}_3 = (1, 0, -1, 0, 0, 0, 0, 0, 0, 0)^{tr}$$

$$w_4(\underline{x}) = \gamma x_2 \qquad , \quad \underline{r}_4 = (0, 1, 0, -1, 0, 0, 0, 0, 0, -1)^{tr}$$

$$w_5(\underline{x}) = (1 - \xi)\gamma x_3 \qquad , \quad \underline{r}_5 = (0, 0, 1, -1, 0, 0, 0, 0, 0, 0)^{tr}$$

$$w_6(\underline{x}) = \gamma x_5 \qquad , \quad \underline{r}_6 = (0, 0, 0, -1, 1, 0, 0, 0, 0, -1)^{tr} \qquad (2)$$

$$w_7(\underline{x}) = \eta v\beta x_1(x_2 + \phi x_3 + \varrho) \qquad , \quad \underline{r}_7 = (1, 0, 0, 0, -1, -1, 0, -1, 0, 0)^{tr}$$

$$w_8(\underline{x}) = \mu x_2 \qquad , \quad \underline{r}_8 = (0, 1, 0, 0, 0, 0, 0, 0, -1, 0)^{tr}$$

$$w_9(\underline{x}) = \mu x_5 \qquad , \quad \underline{r}_9 = (0, 0, 0, 0, 1, 0, 0, 0, -1, 0)^{tr}$$

$$w_{10}(\underline{x}) = \xi\gamma x_3 \qquad , \quad \underline{r}_{10} = (0, 0, 1, -1, 0, 0, 0, 0, 0, -1)^{tr} \quad .$$

With these $w_j(\underline{x})$ and $\underline{r}j$ specified we also can express the mean field ODE system.

The deterministic version of the refined model is given by a differential equation system for all classes, including the recording classes of cumulative cases $C_H$, $C_A$, $C_R$ and $C_U$ by

$$\frac{d}{dt}S = -\beta \frac{S}{N}(H + \phi A + \varrho N)$$

$$\frac{d}{dt}H = \eta(1 - v)\beta \frac{S}{N}(H + \phi A + \varrho N) - (\gamma + \mu)H$$

$$\frac{d}{dt}A = (1 - \eta)\beta \frac{S}{N}(H + \phi A + \varrho N) - \gamma A$$

$$\frac{d}{dt}R = \gamma(H + U + A)$$

$$\frac{d}{dt}U = v\eta\beta \frac{S}{N}(H + \phi A + \varrho N) - (\gamma + \mu)U \qquad (3)$$

$$\frac{d}{dt}C_H = \eta(1 - v)\beta \frac{S}{N}(H + \phi A + \varrho N)$$

$$\frac{d}{dt}C_A = \xi \cdot (1 - \eta)\beta \frac{S}{N}(H + \phi A + \varrho N)$$

$$\frac{d}{dt}C_R = \gamma(H + U + \xi A)$$

$$\frac{d}{dt}C_U = v\eta\beta \frac{S}{N}(H + \phi A + \varrho N)$$

$$\frac{d}{dt}D = \mu(H + U)$$

Model parameters and initial conditions are shown in Table 1, where $\beta$ is the infection rate and $\phi$ is the ratio describing the asymptomatic/mild infections contribution to the force of infection. $\gamma$ is the recovery rate, $\mu$ is the disease induced death rate and $v$ is the ratio of hospitalized going to the ICU. $\eta$ is the proportion of susceptible being infected, develop severe symptoms and being hospitalized whereas $1 - \eta$ is the proportion of susceptible becoming infected

**Table 1. Model parameters and initial condition values.**

| Parameters, variables and initial conditions | Description | Values |
|---|---|---|
| $N$ | population size | $2.2 \times 10^6$ |
| $H(t_0)$ | severe disease and hospitalized | 54.0 |
| $A(t_0)$ | mild disease and asymptomatic | 80.0 |
| $U(t_0)$ | ICU patients | 10.0 |
| $R(t_0)$ | recovered | 1.0 |
| $C_H(t_0)$ | recorded $H(t_0)$ | 54.0 |
| $C_A(t_0)$ | recorded $A(t_0)$ | 40.0 |
| $C_U(t_0)$ | recorded $U(t_0)$ | 10.0 |
| $C_R(t_0)$ | recorded $R(t_0)$ | 1.0 |
| $D(t_0)$ | death | 1.0 |
| $\beta$ | infection rate | $3.25 \cdot \gamma$ |
| $\phi$ | ratio of mild/asymptomatic infections contributing to force of infection | 1.65 |
| $\gamma$ | recovery rate | $0.05 d^{-1}$ |
| $\mu$ | disease induced death rate | $0.02 d^{-1}$ |
| $\nu$ | hospitalized to ICU rate | 0.1 |
| $\eta$ | proportion of hospitalization | 0.4 |
| $\xi$ | detection ratio of mild/asymptomatic | 0.4 |
| $\varrho$ | import parameter | – |

and developing mild disease or asymptomatic. $\xi$ is the ratio of detected, via testing, mild/asymptomatic infect individuals. $\varrho$ is the import rate needed to describe the introductory phase of the epidemics and for the present study, we assume $\varrho$ to be much smaller than the other additive terms of the force of infection, given the strong observational insecurities on the data collected at the beginning of the outbreak.

Model parameters were estimated and fixed as the model is able to describe the disease incidence during the exponential phase of the outbreak. Parameter values were kept close to the ones found in the literature [12–17], used as baseline for initial values and then refined using the available data via the analysis of the likelihood functions [11]. The ratio of mild/asymptomatic infection contribution to the force of infection is estimated to be higher than the contribution of severe hospitalized cases since they would be more mobile and therefore able to transmit the disease while severe cases would be under strict quarantine once detected and admitted to a hospital. We model the effect of transmission of the overall hospitalized individuals and before the hospital admission is notified, symptomatic individuals would contribute significantly to the force of infection. The same assumption is valid for the mild/asymptomatic individuals.

In good agreement, the refined model can describe the hospitalizations, the ICU admissions and the deceased cases, well matched within the median of the 200 stochastic realizations (see Fig 1). The cumulative incidence for all positive cases follows the higher stochastic realizations range and the deviation observed between model simulations and data can be explained by the increasing testing capacities since March 22, 2020, followed by the introduction of rapid tests (blue dots), mostly used as screening tool in nursing homes and public health workers.

## 0.2 Growth rate

After an introductory phase, the epidemic entered into an exponential growth phase, which started in the Basque Country around the March 10, 2020 and due to the effects of the

imposed control measures has left to a slower growth around March 27, 2020 [11]. This exponential growth phase is typical for any outbreak with disease spreading in a completely susceptible population, as observed already in the SIR-system, from the dynamics of the infected $\frac{dI}{dt} = \left( \beta \frac{S}{N} - \gamma \right) \cdot I$ when $S(t) \approx N$, such that a linear differential equation $\frac{dI}{dt} = (\beta - \gamma) \cdot I =: \lambda \cdot I$ with an exponential growth factor $\lambda$ is obtained. This growth factor then can be measured again from disease data via $\lambda = \frac{1}{I} \cdot \frac{dI}{dt} = \frac{d}{dt} \, ln(I)$ giving a straight line in a semi-logarithmic plot of the data.

For larger compartmental models we obtain similarly an exponential growth factor. For the basic SHARUCD model [11] we have the active disease classes $H$ and $A$ with the dynamics given by

$$\frac{d}{dt} \begin{pmatrix} H \\ A \end{pmatrix} = \left[ \begin{pmatrix} \eta \beta \frac{S}{N} & \phi \eta \beta \frac{S}{N} \\ (1-\eta) \beta \frac{S}{N} & \phi (1-\eta) \beta \frac{S}{N} \end{pmatrix} - \begin{pmatrix} (\gamma + \mu + v) & 0 \\ 0 & \gamma \end{pmatrix} \right] \cdot \begin{pmatrix} H \\ A \end{pmatrix} \tag{4}$$

now including disease induced transition to death via the mortality rate $\mu$ and transition to ICU admission with admission rate $v$. For an epidemic in its initial phase, i.e. $S(t) \approx N$, we now have constant matrices $B = \beta \begin{pmatrix} \eta & \phi \eta \\ (1-\eta) & \phi(1-\eta) \end{pmatrix}$ for entries into the disease classes and $G = \begin{pmatrix} (\gamma + \mu + v) & 0 \\ 0 & \gamma \end{pmatrix}$ for exits from the disease classes, where we had infection rate $\beta$ and recovery rate $\gamma$ in the SIR case. With $\underline{x} = (H, A)^{tr}$ we now have with $J = B - G$ the dynamics $\frac{d}{dt} \underline{x} = J \underline{x}$ and its solution

$$\underline{x}(t) = T e^{\Lambda(t-t_0)} T^{-1} \underline{x}(t_0) \tag{5}$$

with matrix exponential including the eigenvalue matrix $\Lambda$ and the transformation matrix $T$ from the eigenvectors of matrix $J$. The eigenvalues of the matrix $J$ are given by

$$\lambda_{1/2} = \frac{1}{2} \cdot tr \pm \sqrt{\frac{1}{4} \cdot tr^2 - det} \tag{6}$$

with the parameter dependent trace $tr = (\eta + \phi(1 - \eta)) \cdot \beta - (2\gamma + \mu + v)$ and determinant $det = \gamma(\gamma + \mu + v) - ((\gamma + \mu + v)\phi(1 - \eta) + \gamma\eta) \cdot \beta$ and the dominating growth factor is given by the largest eigenvalue $\lambda_1$. After an initial introductory phase the exponential growth with $\lambda_1$ dominates the dynamics of $H(t)$ and $A(t)$, and from there also all the other variables, because the remaining equations are all inhomogeneous linear differential equations with the inhomogeneities given by the solutions $H(t)$ and $A(t)$, and we have

$$H(t) \to K_H \cdot e^{\lambda_1(t-t_0)}, \qquad A(t) \to K_A \cdot e^{\lambda_1(t-t_0)} \tag{7}$$

with constants $K_H$ and $K_A$ depending on parameters and initial conditions (from $H(t) = K_{H,1} \cdot e^{\lambda_1}(t - t_0) + K_{H,2} \cdot e^{\lambda_2}(t - t_0)$ etc.). In the limiting case of a simple SIR-type model (with $\phi \approx 1$ and $\mu, v \ll \gamma$) we obtain $tr = (\eta + \phi(1 - \eta)) \cdot \beta - (2\gamma + \mu + v) \approx \beta - 2\gamma$ and $det = \gamma(\gamma + \mu + v) - ((\gamma + \mu + v)\phi(1 - \eta) + \gamma\eta) \cdot \beta \approx \gamma^2 - \gamma\beta$ and hence $\lambda_1 \approx \beta - \gamma$ and $\lambda_2 \approx -\gamma$.

The concept of the growth rate can be extended into the phase when effects of the control measures become visible and parameters slowly change, such that for short times the above analysis holds as for constant parameters. The momentary growth rates are analyzed below.

## 0.3 Reproduction ratio

Another measure of the spreading of the disease in its initial phase is the basic reproduction number ($R_0$), the number of secondary cases $I_s$ from a primary case $I_p$ during its infectiveness before recovering in a completely susceptible population.

In its simplest version for SIR models a primary case, $I_p(t_0) = 1$, recovers via $\frac{dI_p}{dt} = -\gamma I_p$, hence $I_p(t) = I_p(t_0)e^{-\gamma(t-t_0)}$. The number of secondary cases from the primary case is given by $\frac{dI_s}{dt} = \beta \frac{S}{N} I_p(t)$ with $I_s(t_0) = 0$, a simple inhomogeneous linear differential equation in case of a entirely susceptible population $S(t) = N$. The solution is $I_s(t) = \frac{\beta}{\gamma} \cdot I_p(t_0)(1 - e^{-\gamma(t-t_0)}) + I_s(t_0)$ and gives the total number of secondary cases from a primary case as the long time limit as $I_s(t \to \infty) = \frac{\beta}{\gamma}$, hence the basic reproduction number is simply $\mathcal{R}_0 = \frac{\beta}{\gamma}$. So we have the relation between $\mathcal{R}_0$ and the growth rate $\lambda$ here as $\mathcal{R}_0 = \frac{\beta}{\gamma} = 1 + \frac{\lambda}{\gamma}$. Generalized, the reproduction ratio is then given by $r = I_s(t \to \infty)/I_p(t_0) = \frac{\beta}{\gamma}$ as the ratio of secondary cases produced by primary cases during their infectiousness.

This concept can be also generalized for larger compartmental models, with the notions of matrices $B$ and $G$ as introduced above. For any primary cases $H_p$ or $A_p$ we have with $\underline{x}p = (H_p, A_p)^{tr}$ the decay dynamics $\frac{d}{dt}\underline{x}p = -G \cdot \underline{x}p$ with solution $\underline{x}p(t) = e^{-G(t-t_0)} \cdot \underline{x}p(t_0)$, using again the matrix exponential. For secondary cases $H_s$ and $A_s$ we have the dynamics of $\underline{x}s = (H_s, A_s)^{tr}$ given by $\frac{d}{dt}\underline{x}s = B \cdot \underline{x}p(t)$ with solution analogously to the SIR case as $\underline{x}s(t) - \underline{x}s(t_0) = BG^{-1}(1 - e^{-G(t-t_0)})\underline{x}p(t_0)$ with $F = BG^{-1}$ the next generation matrix, since $\underline{x}s(t \to \infty) = F\underline{x}p(t_0)$ or from generation $\underline{x}n$ to generation $\underline{x}n + 1$ the discrete iteration $\underline{x}n + 1 = F \cdot \underline{x}n$. For the present case we have the next generation matrix given as

$$F = \begin{pmatrix} \dfrac{\eta\beta}{\gamma + \mu + \nu} & \dfrac{\phi\eta\beta}{\gamma} \\[2ex] \dfrac{(1-\eta)\beta}{\gamma + \mu + \nu} & \dfrac{\phi(1-\eta)\beta}{\gamma} \end{pmatrix} \tag{8}$$

with its dominant eigenvalue for the basic SHARUCD-model

$$r_1 = \frac{\eta\gamma + (1-\eta)\phi(\gamma + \mu + \nu)}{\gamma(\gamma + \mu + \nu)} \cdot \beta \tag{9}$$

and the other one being zero. In the limiting case of a simple SIR-type model (with $\phi \approx 1$ and $\mu, \nu \ll \gamma$) we obtain again $r_1 = \beta/\gamma$ as can be easily seen.

This concept of the reproduction ratio can be extended into the phase when effects of the control measures become visible and parameters slowly change. The momentary reproduction ratios ($r$) can be analyzed, as frequently done for the COVID-19 epidemics, but often called "basic reproduction number". While the momentary growth rate follows directly from the time continuous data at hand, the momentary reproduction ratio depends on the notion of a generation time $\gamma^{-1}$.

To obtain the momentary growth rates from data directly we use $\lambda = \frac{d}{dt} \ln(I)$ at first applied to the cumulative tested positive cases $I_{cum}(t)$ obtaining, via a smoothing window, the new cases after time $\tau$ as

$$I_{new,\tau}(t) := I_{cum}(t) - I_{cum}(t - \tau) \tag{10}$$

and hence, the growth rate

$$\lambda = \frac{1}{\Delta t}\left( ln(I_{new,\tau}(t)) - ln(I_{new,\tau}(t - \Delta t)) \right) \quad . \tag{11}$$

## 1 Results and discussion

From the growth rate, the reproduction ratio is calculated with the recovery period $\gamma^{-1}$ obtained from our underlying model and recent literature about SARS-CoV-2 interaction with human hosts [16–21]. Assuming the recovery period to be of 10 days, we use 7 days smoothing of the differences of logarithmic positive cases to include all possible fluctuations during the data collection process such as "weekend effects", for example, when we often observe a consistent low number of cases reported that are then adjusted shortly after. For this first exercise, both data sets show negative growth rates from April 1st, 2020, confirming a decrease in disease transmission. Nevertheless, when looking at the long term results for the "Data set A" (see Fig 2a and 2b)), an increase of the growth rate over time is estimated, with values crossing the threshold and becoming positive from April 23 to May 1, 2020, whereas "Data set B" (see Fig 2c and 2d), measures were kept constantly negative, without any signal of increasing disease transmission. The momentary reproduction numbers follow the same trends for both data sets respectively, depending on the data set used. The observed signal from "Data set A" would significantly impact decisions on lockdown lifting, as the national plan for lifting the restrictions imposed during the state of alarm, called "Plan for the Transition towards a new normality", was announced on April 28, 2020 [22]. Taking place over 4 phases, with a gradual de-escalation to "a new normality", the plan is dependent on the ongoing progress of COVID-19 epidemic's control across the different regions of Spain. However, results obtained by "Data set B" would, alternatively, support the already started lockdown lifting with its "Phase Zero" initiated on May 4, 2020. When assuming a short recovery period of $\gamma$ = 4 days, see Fig 4, similar results are observed between the different data sets, only with variation on the absolute values.

The momentary growth rates for the various variables are also calculated to verify and support the interpretation of the estimated $r(t)$ threshold behaviour since for any assumed recovery period $\gamma^{-1}$, results obtained for the various variables are the same, changing only when considering the different data sets. Fig 4 shows the behavior of three variables that are synchronized in the Basque Country, $I_{cum}$, $C_H$, and $C_U$. They also cross the threshold to a negative growth rate on April 1st, 2020, confirming the observed $r(t)$ trend obtained by looking at data on $I_{cum}$ alone. Recovered and deceased cases, shown in Fig 4b), follow 1-2 weeks later, due to the delay between onset of symptoms, hospitalization and eventually death, reaching negative growth rate on April 7 and April 11, 2020, respectively. Besides the observed deviation of $I$, for the "Data set A", the other variables are kept below the threshold, constantly negative until May 9, 2020, supporting the political decision of starting lifting the lockdown measures rather sooner than later in time. So which measure should be considered to guide political decisions? Here, the answer is simple. When the available data is consistently collected and defined, the momentary growth rates for different variables, $I$, $H$, $U$, $R$ and $D$, measured directly from the data at hand, should also be considered as complementary investigation.

## Conclusion

As the concept of $R_0$ used alone can be easily misinterpreted, specially now when testing capacity is increasing and consequently the number of new notified cases, the BMTF now monitors

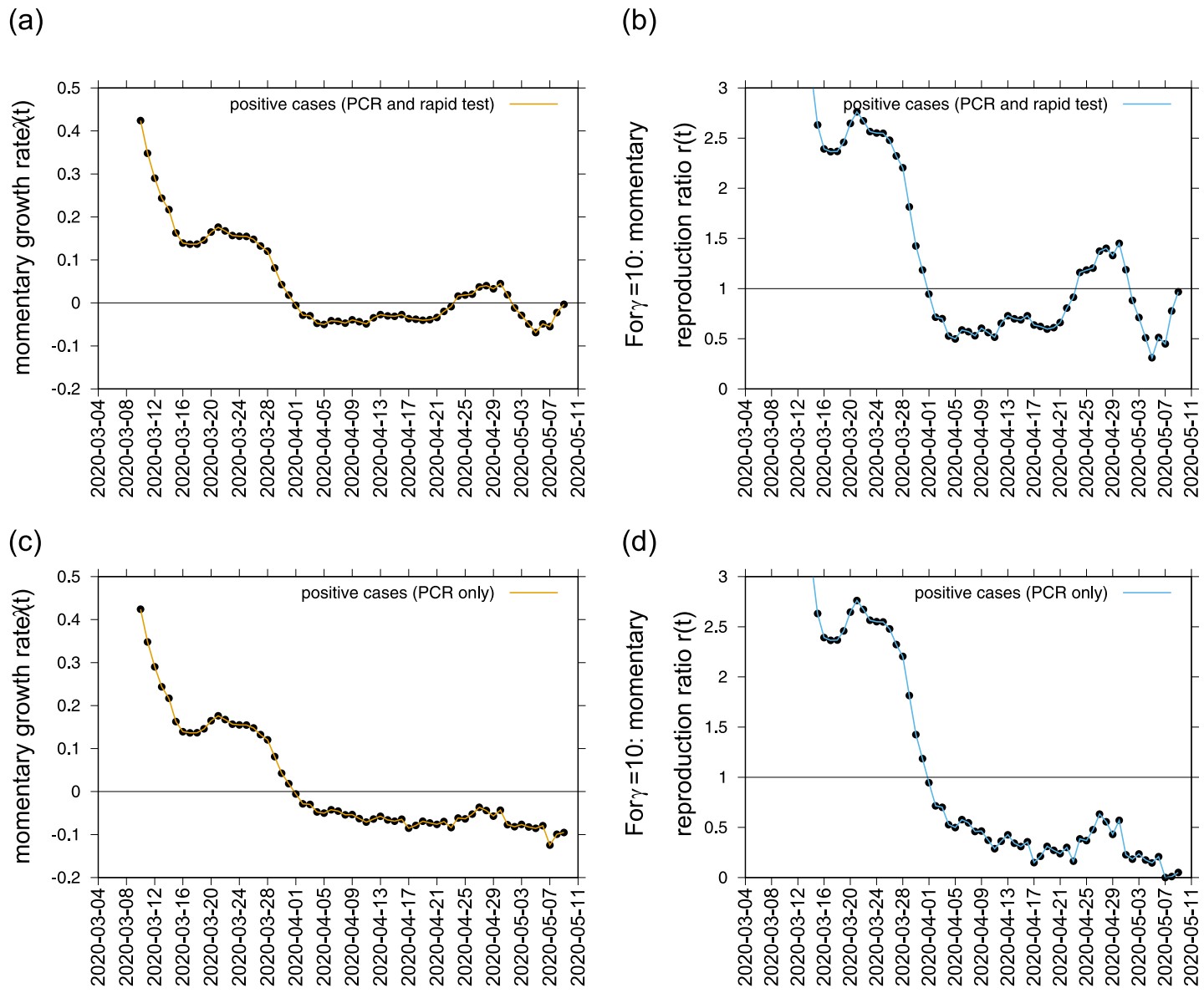

**Fig 2.** Momentary growth rates estimation from the data on positive tested infected cases in a) PCR and rapid tests and c) PCR alone. The momentary reproduction ratios from the same data respectively are shown in b) and d), for $\gamma^{-1} = 10$.

the development of the COVID-19 epidemic in the Basque Country by considering not only the behaviour of the momentary growth rates $\lambda(t)$ and momentary reproduction numbers $r(t)$ for the positive cases $I_{cum}(t)$, but also the $\lambda(t)$ for hospitalizations ($C_H$), ICU admissions ($C_U$), deceased ($D$) and recovered cases ($C_R$), assisting the Basque Health Managers and the Basque Government with results that are obtained by the model framework, based on available data and evidence as scientific advise. Without interfering in any political decision, we now use "Data set B", with a clearer definition of tested positive cases $I_{cum}$ and all other variables that follow, $C_H$, $C_U$, $C_R$ and $D$, and recovery period of $\gamma = 4$, shown in Figs 3d, 4c and 4d. At the moment, the reproduction ratio $r$ is estimated to be below the threshold behavior of $r = 1$,

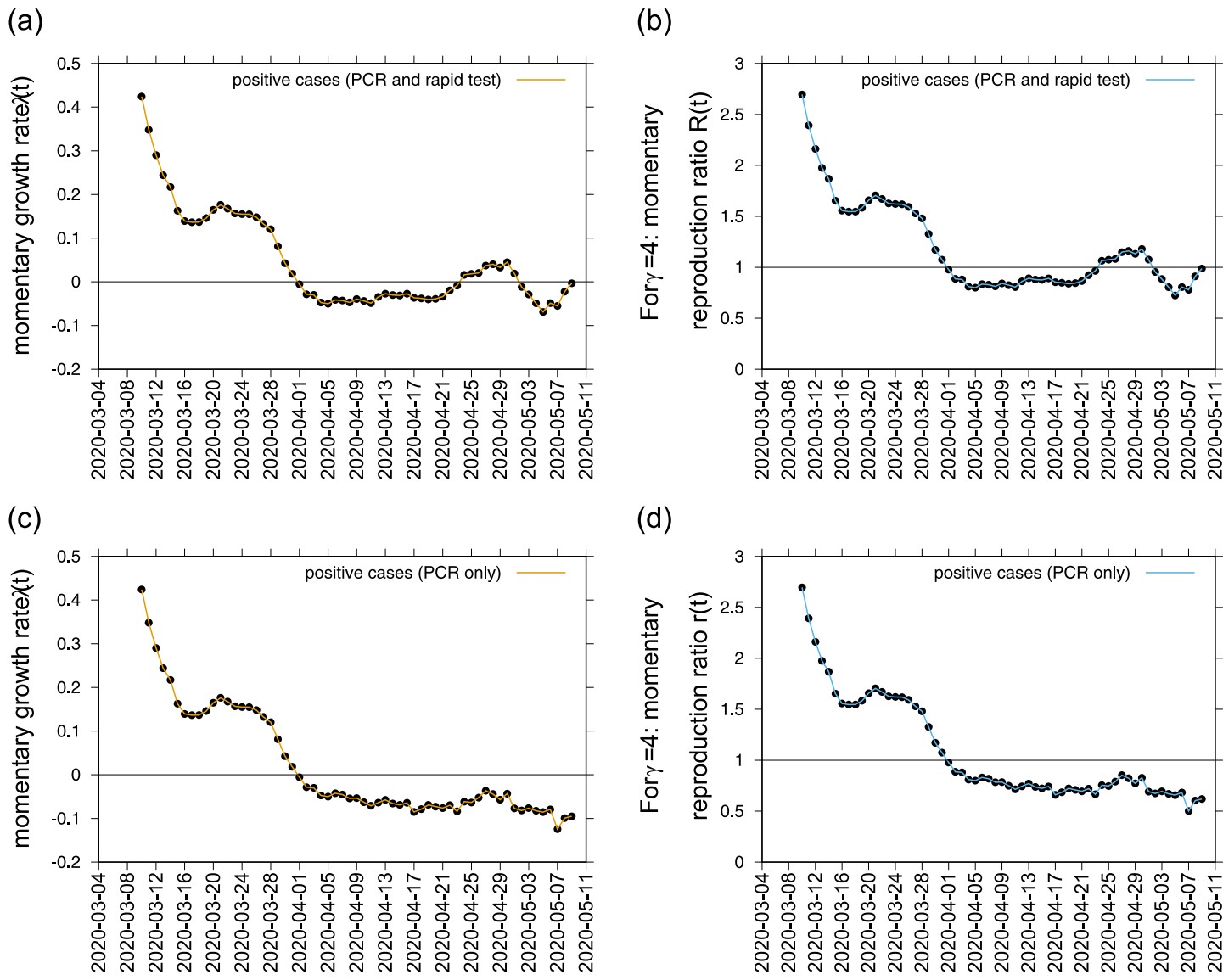

**Fig 3.** Momentary growth rates estimation from the data on positive tested infected cases in a) PCR and rapid tests and c) PCR alone. The momentary reproduction ratios from the same data respectively are shown in b) and d) for $\gamma^{-1} = 4$.

but still close to 1, meaning that although the number of new cases reported in the Basque Country are decelerating, the outbreak is still in its linear phase and careful monitoring of the development of the dynamics of the new cases from all categories, based on new information and data, to support the upcoming political decisions that will change the current life situation of millions of people is required.

Using the available data for the Basque Country, a small community with short path for data collection and validation, we developed a modeling framework able to predict the course of the epidemic, from introduction to control measure response, potentially helpful for task forces around the globe which are now struggling to provide real scientific advice for health managers and governments while the lockdown measures are relaxed.

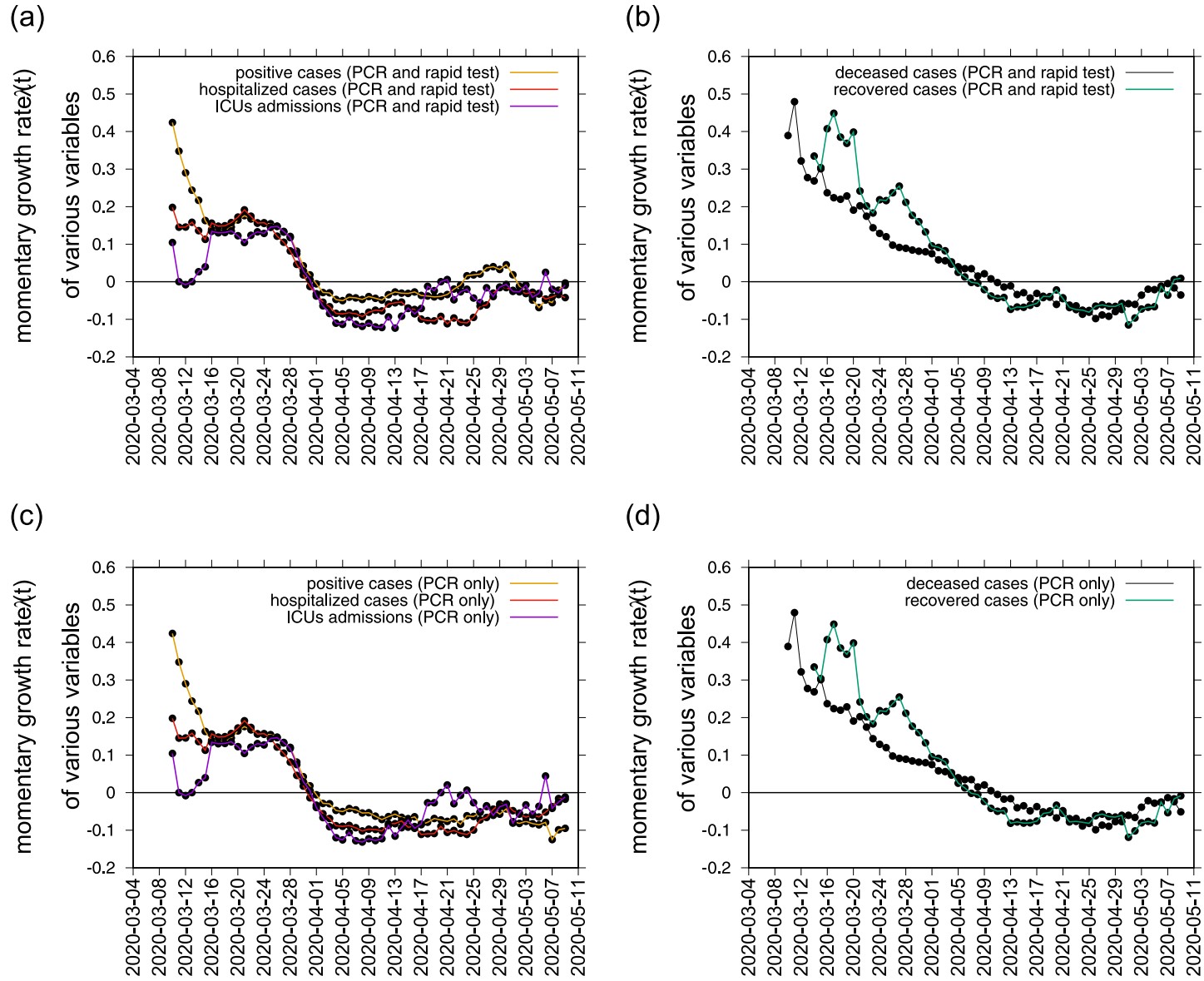

**Fig 4.** Using data on PCR and rapid tests we plot the momentary growth rates estimation from the data on positive tested infected cases (yellow), hospitalizations (red) and ICU admission (purple) are plotted in a) and recovered (green) and deceased cases (black) in b). Using data on PCR tests only we plot the momentary growth rates estimation from the data on positive tested infected cases (yellow), hospitalizations (red) and ICU admission (purple) are plotted in c) and recovered (green) and deceased cases (black) in d).

## Acknowledgments

We thank the huge efforts of the whole COVID-19 BMTF, specially to Eduardo Millán for collecting and preparing the data sets used in this study. We thank Adolfo Morais Ezquerro, Vice Minister of Universities and Research of the Basque Goverment for the fruitful discussions.

## Author Contributions

**Conceptualization:** Maíra Aguiar, Nico Stollenwerk.

**Formal analysis:** Maíra Aguiar, Joseba Bidaurrazaga Van-Dierdonck, Nico Stollenwerk.

**Investigation:** Maíra Aguiar, Joseba Bidaurrazaga Van-Dierdonck.

**Writing – original draft:** Maíra Aguiar, Joseba Bidaurrazaga Van-Dierdonck.

**Writing – review & editing:** Maíra Aguiar, Nico Stollenwerk.

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
