## [Decision Letter · Decision Letter 0]

25 Jun 2020

PONE-D-20-14985

Reproduction ratio and growth rates: measures for an unfolding pandemic

PLOS ONE

Dear Dr. Aguiar,

Thank you for submitting your manuscript to PLOS ONE. After careful consideration, we feel that it has merit but does not fully meet PLOS ONE’s publication criteria as it currently stands. Therefore, we invite you to submit a revised version of the manuscript that addresses the points raised during the review process.

In the revised version, please also provide  figures with a better resolution of the size of the fonts and labels are too small.

We look forward to receiving your revised manuscript.

Kind regards,

Constantinos Siettos, Ph.D.

Academic Editor

PLOS ONE

Journal Requirements:

Additional Editor Comments (if provided):

Reviewers' comments:

Reviewer's Responses to Questions

**Comments to the Author**

1. Is the manuscript technically sound, and do the data support the conclusions?

Reviewer #1: Yes

2. Has the statistical analysis been performed appropriately and rigorously? 

Reviewer #1: Yes

3. Have the authors made all data underlying the findings in their manuscript fully available?

Reviewer #1: Yes

4. Is the manuscript presented in an intelligible fashion and written in standard English?

Reviewer #1: No

5. Review Comments to the Author

Reviewer #1: see attached report

6. PLOS authors have the option to publish the peer review history of their article (what does this mean?). If published, this will include your full peer review and any attached files.

Reviewer #1: No

---

## [Author Response · Author response to Decision Letter 0]

29 Jun 2020

Dear Editor and Reviewer of PloS One Journal,

Thank you for your comments and for allowing us to revise our manuscript. In the revised version we addressed your comments, criticisms and suggestions.

We thank the referee for carefully reading our paper and for the polite tone of making comments. We really appreciated that. A detailed reply on the comments of the reviewer was uploaded separately.

---

## [Editor Report · Decision Letter 1]

13 Jul 2020

Reproduction ratio and growth rates: measures for an unfolding pandemic

PONE-D-20-14985R1

Dear Dr. Aguiar,

We’re pleased to inform you that your manuscript has been judged scientifically suitable for publication and will be formally accepted for publication once it meets all outstanding technical requirements.

Kind regards,

Constantinos Siettos, Ph.D.

Academic Editor

PLOS ONE
---

## [Editor Report · Acceptance letter]

15 Jul 2020

PONE-D-20-14985R1 

Reproduction ratio and growth rates: measures for an unfolding pandemic 

Dear Dr. Aguiar:

I'm pleased to inform you that your manuscript has been deemed suitable for publication in PLOS ONE. Congratulations! Your manuscript is now with our production department. 

Kind regards, 

on behalf of

Professor Constantinos Siettos 

Academic Editor

PLOS ONE